# Momentarily trapped exciton polaron in two-dimensional lead halide perovskites

Weijian Tao [1], Chi Zhang[1], Qiaohui Zhou[1], Yida Zhao[1] & Haiming Zhu [1✉]

Two-dimensional (2D) lead halide perovskites with distinct excitonic feature have shown exciting potential for optoelectronic applications. Compared to their three-dimensional counterparts with large polaron character, how the interplay between long- and short- range exciton-phonon interaction due to polar and soft lattice define the excitons in 2D perovskites is yet to be revealed. Here, we seek to understand the nature of excitons in 2D CsPbBr$_3$ perovskites by static and time-resolved spectroscopy which is further rationalized with Urbach-Martienssen rule. We show quantitatively an intermediate exciton-phonon coupling in 2D CsPbBr$_3$ where exciton polarons are momentarily self-trapped by lattice vibrations. The 0.25 ps ultrafast interconversion between free and self-trapped exciton polaron with a barrier of ~ 34 meV gives rise to intrinsic asymmetric photoluminescence with a low energy tail at room temperature. This study reveals a complex and dynamic picture of exciton polarons in 2D perovskites and emphasizes the importance to regulate exciton-phonon coupling.

---

[1] State Key Laboratory of Modern Optical Instrumentation, Centre for Chemistry of High-Performance & Novel Materials, Department of Chemistry, Zhejiang University, Hangzhou Zhejiang, China. ✉email: hmzhu@zju.edu.cn

 1

Recently, two-dimensional (2D) Ruddlesden–Popper (RP) phase lead halide perovskites have emerged as promising light-emitting materials due to their solution processing, tunable bandgap, and high photoluminescence (PL) quantum yield (QY)[1–4]. With large quantum confinement and reduced dielectric screening, excitons in 2D perovskites are strongly confined with large exciton binding energy[5,6]. Different from conventional covalent semiconductors (e.g. Si, GaAs), the soft and ionic lattice of lead halide perovskites exhibits highly anharmonic fluctuations[7,8] and is deformable under photoexcitation[9,10]. As a result, exciton–phonon coupling dresses excitons in 2D perovskite with significant polaronic character[11,12], leading to complex excited-state energetic/configurational landscape[13–15].

Depending on the nature of exciton (carrier)–phonon interaction, exciton (carrier) in lead halide perovskites can exist as exciton polaron (large polaron) which can move freely but dressed with phonon clouds through long-range Frohlich-like interaction, or self-trapped exciton (STE) (small polaron) through short-range Holstein-like interaction[12,16]. While free exciton (FE) polarons can move coherently and are characterized with relatively narrow and near-band edge emission[17], STEs are tightly localized by lattice distortion and typically exhibit a broad emission with large Stokes shift[18]. In fact, large polaron and STE have been established in photovoltaic three-dimensional (3D) perovskites[9,10,19] and white light-emitting 2D perovskites with corrugated lattice[20], respectively. However, excitons in 2D RP perovskites remain poorly defined so far. The coexistence and complex interplay between long- and short-range interactions put 2D RP perovskites in a unique intermediate regime and impose significant challenges on understanding and describing their exciton behaviors[12].

Here we show that excitons in 2D lead bromide RP perovskites are neither FE polarons nor localized by extrinsic disorders/ defects or intrinsic self-trapping effect, but instead, momentarily trapped by lattice fluctuations. By PL measurements, we observed an asymmetric PL line shape in two-layer (2L) CsPbBr$_3$ nanoplates (NPs) with a dominant FE polaron peak and a thermally populated low-energy exponential tail corresponding to STE. The STE reaches thermal equilibrium with FE polaron in ~0.4 ps at room temperature with a small activation energy of ~34 meV. These results were rationalized with Urbach–Martienssen rule pointing to intermediate exciton–phonon coupling strength and metastable STEs in 2D RP perovskites. The momentarily trapped exciton polarons in 2D perovskites have strong implications to their optoelectronic applications.

## Results

**Characterization of 2L CsPbBr$_3$ NPs.** Two-layer CsPbBr$_3$ perovskite NPs were synthesized according to literature[21] with a slight modification (see "Methods" for details). The NPs are capped with oleylaminium (OLA) ligands and can be well dispersed in nonpolar solvents or deposited into films using drop cast or spin coating. Compared to conventional 2D RP perovskites with butylammonium (BA) or phenylethylammonium (PEA) ligands, we also call these NPs as 2D RP perovskites because of same inorganic lattice, ligand binding group, and optical properties. In fact, this kind of 2D perovskite NPs can be reversibly assembled into ordered superstructure, i.e. conventional multilayered RP perovskite[22]. From the transmission electron microscope (TEM) image (Fig. 1a), the NPs have a lateral size of 20 ± 3 nm and a thickness of <2 nm. As these atomically thin perovskite NPs are highly susceptible to electron beam damage, we turn to optical measurements to characterize their exact thickness and homogeneity, which provides a facile and precise approach[6]. The absorption spectrum (Fig. 1b) exhibits a strong and sharp

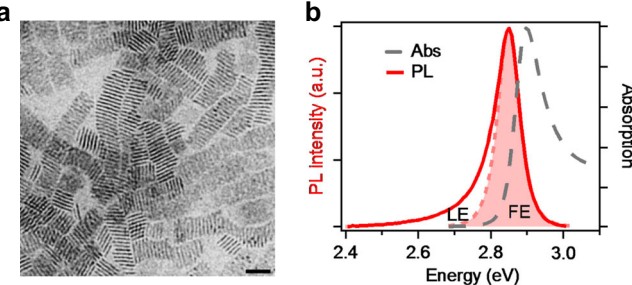

**Fig. 1 Characterization of 2L CsPbBr$_3$ NPs. a** TEM image of 2L CsPbBr$_3$ NPs. Scale bar is 20 nm. **b** Absorption and PL spectra of 2L CsPbBr3 NPs in toluene solution. The red shaded area is vogit fitting of main peak, which is denoted as FE and the rest as LE.

peak at 2.90 eV, corresponding to the lowest energy excitonic transition of 2L CsPbBr$_3$ perovskite due to quantum confinement effect[5,6]. The PL spectrum shows a distinct peak at 2.85 eV. The PL QY is measured to be ~38%, which is comparable with the highest reported PL QY (49%) of 2L CsPbBr$_3$ (ref. [6]) and suggests high sample quality. The absence of other PL peaks confirms the thickness homogeneity. The ~50 meV PL Stokes shift is considerably larger than that of II–VI group colloidal NPs (e.g. CdSe) with similar thickness (<10 meV)[23,24]. We attribute this to the inherent polaronic character of the exciton in lead halide perovskite ionic lattice[25]. Interestingly, the PL spectrum shows an asymmetric line shape with a low-energy tail extending to ~450 meV below the peak. This peak asymmetry can be better viewed in logarithm plot (Supplementary Fig. 1) which shows clearly an exponential tail. This asymmetric PL spectra have been observed in 2D perovskites of different forms (single crystals[26], polycrystalline films[27] and, NPs[21]), suggesting its intrinsic origin. We fit the main peak with a symmetric vogit line shape and denoted it as FE emission since it is sharp and close to the absorption peak and the remaining low-energy tail as localized exciton (LE) emission as it is quite broad and lower in energy. The exact origin about the LE emission, e.g. whether it is intrinsic or extrinsic, is unclear yet.

**PL study of 2L CsPbBr3 NPs.** Photoluminescence excitation spectrum (PLE) was applied to get a first glance of the origin of the low-energy tail. The PLE spectra monitored at 2.92 eV and 2.58 eV are displayed in Fig. 2a, corresponding to the FE and LE emissions, respectively. The identical PLE spectra indicate FE and LE emissions are relaxed from the same excited species and rule out the phase or thickness inhomogeneity. To quantitatively characterize the spectral asymmetry, we introduce two dimensionless parameters, LE ratio and asymmetric factor ($A_S$)[28]:

$$\text{LE ratio} = I_{LE}/I_{FE} \quad (1)$$

$$A_S = W_L/W_H \quad (2)$$

where $I_{LE}$ and $I_{FE}$ are peak areas of LE and FE, $W_L$ and $W_H$ are peak half-widths at 10% of the peak amplitude for the low-energy and high-energy side, respectively. The closer to 0 (1) for LE ratio ($A_S$), the more symmetric the peak is. LE ratio and $A_S$ were calculated to be ~0.4 and ~2.0 for 2L CsPbBr$_3$, respectively, at room temperature. We varied the excitation power by four orders of magnitude and the LE ratio and $A_S$ show no change (Fig. 2b), suggesting the spectral asymmetry with LE emission does not stem from the extrinsic disorder/defects which, otherwise, would be filled at high excitation power.

We also measured PL decay kinetics at different emission energies (from 2.38 to 2.86 eV). Interestingly, regardless of

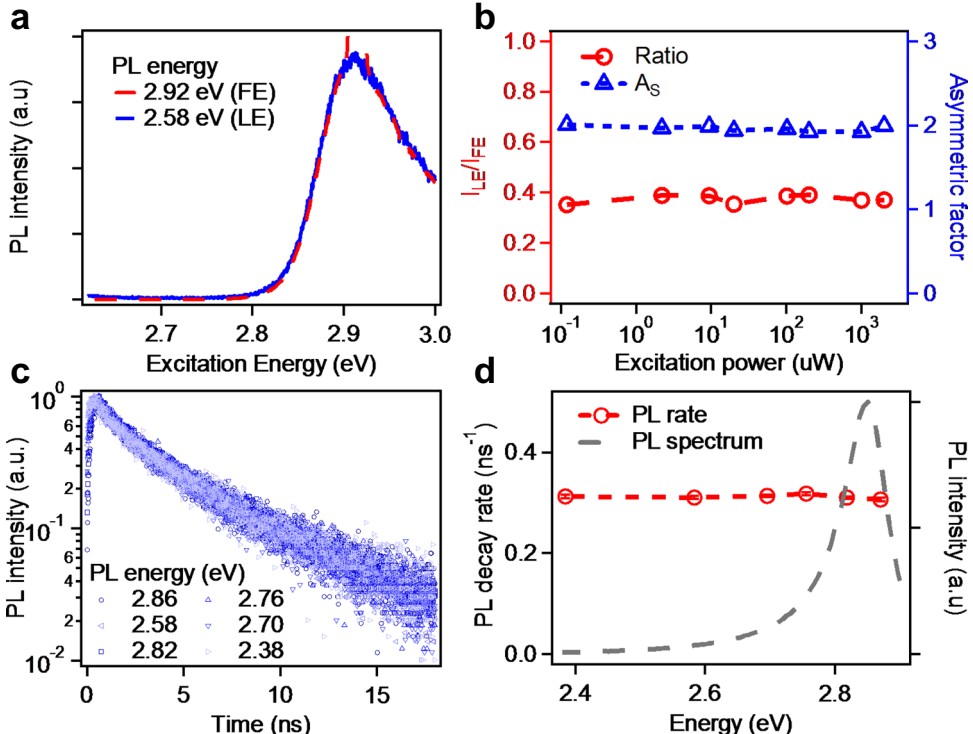

**Fig. 2 Equilibrium of FE and LE emission at room temperature. a** PLE spectra of 2L CsPbBr$_3$ NPs monitored at FE emission (2.92 eV) and LE emission (2.58 eV). **b** LE ratio and asymmetric factor as a function of excitation power. **c** PL decay kinetics and **d** PL decay rate at different energies (2.38–2.86 eV).

emission energy, PL decay shows same kinetics and a constant lifetime of ~3.2 ns at room temperature (Fig. 2c, d). This is in striking contrast to conventional III–V semiconductors[29,30] and lead halide perovskite polycrystalline films[31] which show a longer lifetime at lower emission energy due to more localized character. This is also not expected for surface/edge state emission which was observed previously in 2D perovskites and characterized with lower energy and longer PL lifetime compared to FE emission[32]. The energy-independent PL decay kinetics suggests thermal equilibrium between FE and LE at room temperature.

**Temperature-dependent PL and transient absorption study**. To unambiguously establish the thermal equilibrium and extract the energy difference between FE and LE states, we performed temperature-dependent PL measurements. The PL spectra at different temperatures and the vogit fits are shown in Fig. 3a (298 K, 230 K, 77 K) and Supplementary Fig. 2 (other temperatures). Interestingly, with decreasing temperature from 298 K to 77 K, PL spectrum becomes not only narrower but also more symmetric. Both LE ratio and $A_S$ decrease with temperature, reaching ~0.1 and 1 below 110 K, respectively, indicating negligible LE emission contribution and symmetric line shape (Fig. 3b). This temperature-dependent behavior of LE emission cannot be ascribed to surface/edge-related states which should yield more pronounced LE emission at lower temperature[33]. The decreasing of LE emission at lower temperature indicates LE state is higher in energy than FE state and is thermally populated at room temperature (inset in Fig. 3c). We estimated the energy difference ($\Delta E = E_{LE} - E_{FE}$) between LE state and FE state from the temperature-dependent PL results. Under thermal equilibrium, the LE/FE ratio obeys[18]

$$\ln \frac{I_{LE}}{I_{FE}} \propto \ln \frac{k_{r,LE}}{k_{r,FE}} - \frac{\Delta E}{k_B T} \qquad (3)$$

where $k_{r,LE}$ and $k_{r,FE}$ are recombination rate constants of LE and

FE, respectively; $k_B$ is Boltzmann constant and $T$ is temperature. The plot of $\ln \frac{I_{LE}}{I_{FE}}$ vs $\frac{1}{T}$ exhibits a linear relationship, from which we extracted a $\Delta E$ of ~34 meV (Fig. 3c). This energy difference is only slightly larger than room temperature thermal energy (~25 meV), indicating a significant population of LE state at room temperature. Based on Boltzmann distribution, LE-state population is estimated to be ~28% of the FE-state population at room temperature, thus the emission from LE state can be observed. We note the relative PL intensity is also affected by the radiative rate constant, which unfortunately cannot be easily determined since they show identical PL decay kinetics under thermal equilibrium (Fig. 2c).

We performed transient absorption (TA) study to follow exciton thermal equilibration process in real time with femtosecond time resolution at room temperature. We excited 2L CsPbBr$_3$ with a low-energy pulse (with a center at 2.75 eV and a high energy cutoff at 2.83 eV) to selectively excite the exponential absorption tail which is dominated by LE or the momentarily localized exciton transition[34] and after a certain delay time, measured the transmission change of a white light continuum. As shown in Fig. 3d, a low-energy bleach peak with maximum at $E_{max}$ ~2.83 eV forms instantly upon photoexcitation, which can be ascribed to LE transition. The TA bleach shifts to higher energy progressively in 0.4 ps, leading to the bleach of FE transition at 2.88 eV. The evolution of $E_{max}$ after photoexcitation is plotted in Fig. 3e, which shows a half-life time of ~0.25 ps for thermal equilibration process. This thermal equilibration time is orders of magnitude faster than exciton recombination process which explains wavelength-independent PL decay kinetics (Fig. 2c).

**Discussion**

We now seek to understand the origin of the asymmetric PL line shape at room temperature and more importantly, the exciton behavior in 2D lead halide perovskites. Low-energy tail in PL

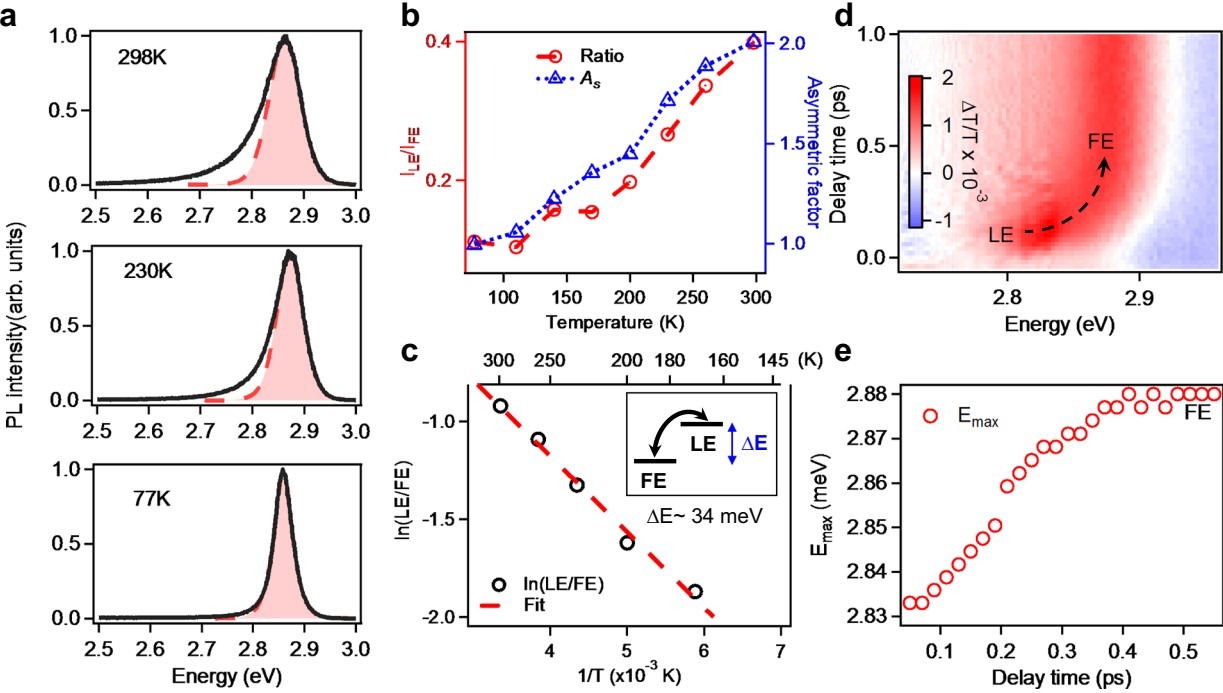

**Fig. 3 Dynamics between FE and LE. a** PL spectra at 298 K, 230 K, and 77 K, respectively, and the vogit fits. **b** LE ratio and asymmetric factor as a function of temperature. **c** Logarithm plot of LE ratio as a function of temperature and the linear fit according to Eq. 3. Inset: scheme showing the energy difference and conversion between FE state and LE state. **d** 2D color plot of TA spectra under tail excitation. **e** The shift of bleach peak maximum $E_{max}$ with pump-probe decay time.

spectra has been generally observed in semiconductor systems such as III–V semiconductor alloys[29,30,35], doped semiconductors[36], carbon nanotubes[37], and ionic/polar solids[38–40]. And several mechanisms have been proposed such as extrinsic structure/impurity traps[36,41], potential fluctuations[29,30,35], phonon replica broadening[40], Fermi-edge singularity effect[37], and Urbach exciton emission[38,39,42]. Previous studies on 2D lead halide perovskites also have shown asymmetric PL line shape and attributed to extrinsic below gap trap states at surfaces/interfaces[33,43]. Layer edge states can also contribute the lower emission due the stochastic loss of organic ligands and formation of bulk CsPbBr3 perovskites[2,32,44–47]. However, lower energy emissions from below gap states or layer edge states exhibit longer lifetime compared with the band edge emission from 2D perovskites and persist at cryogenic temperature, if not be more pronounced[2,32,33,46,48]. Here in 2L CsPbBr$_3$ perovskite, the power-independent spectral shape and ultrafast thermal equilibration with a $\Delta E$ of ~34 meV preclude extrinsic origins such as middle gap states from defects or potential fluctuations from thickness or phase inhomogeneity or unintentionally formed edge states. Otherwise, the low-energy tail extending to ~450 meV below main peak would indicate a wide distribution of deep trap states or edge states with energy-dependent recombination rates due to localization effect and excitons will be localized to these states at low temperature[33], opposite to experimental results. The absence of extrinsic edge/defect-related emission confirms high sample quality, which is key to reveal the intrinsic optical properties.

The intrinsic nature of the asymmetric PL with a low-energy tail can be attributed the polaronic effect in lead halide perovskites and we assign the LE emission to intrinsic STE emission. Carriers (or excitons) in lead halide perovskites can couple to lattice vibrations through both Fröhlich-like long-range interaction, due to their polar and ionic lattice, and Holstein-like short-range

interaction, due to the soft and dynamic nature[7,12,16,25]. The interplay between them determines the excited-state energy landscape. According to polaron theory and phase diagram, a large polaron (or a FE polaron), which moves freely and coherently but is dressed with phonon cloud[10], is the stable species in materials with dominant long-range coupling. This has been the case for 3D bulk lead halide perovskites, leading to the strong and sharp near-band gap emission, long carrier lifetime but modest carrier mobility[49,50]. Further increasing short-range coupling reaches small polaron (or STE) limit where carriers (excitons) induce large lattice distortion and reorganization through short-range interaction. This has been the case for white light emitting 2D layered perovskites with corrugated lead halide octahedral sheets. There, a broad and highly Stokes-shifted (~600 meV) emission has been generally observed and ascribed to intrinsic STE emission[20,51]. Considering similar compositions in 2D RP perovskites with flat Pb–Br sheets, the asymmetric PL emission with both FE and STE contribution indicates the coexistence and dynamic equilibrium between FE polaron and STE, as shown by temperature-dependent and time-resolved studies above. In another word, excitons in these 2D perovskites are momentarily localized, varying from time to time following lattice vibrations[52]. The small energy difference between FE polaron and STE in 2L CsPbBr$_3$ indicates that the exciton–phonon coupling is in an intermediate regime between Fröhlich-like exciton polaron limit and Holstein-like STE limit.

Here, we follow the model proposed by Toyozawa[52,53] to quantitatively elucidate the underlying exciton–phonon interaction strength and exciton behavior in 2D CsPbBr$_3$ perovskite. In high-quality semiconductors with less defects and extrinsic disorders, as exciton is influenced by the phonon field, exciton–phonon interaction is the main reason for the low-energy exponential tail of the absorption peak, which is often called Urbach–Martienssen tail (U–M tail)[52]. According to this model,

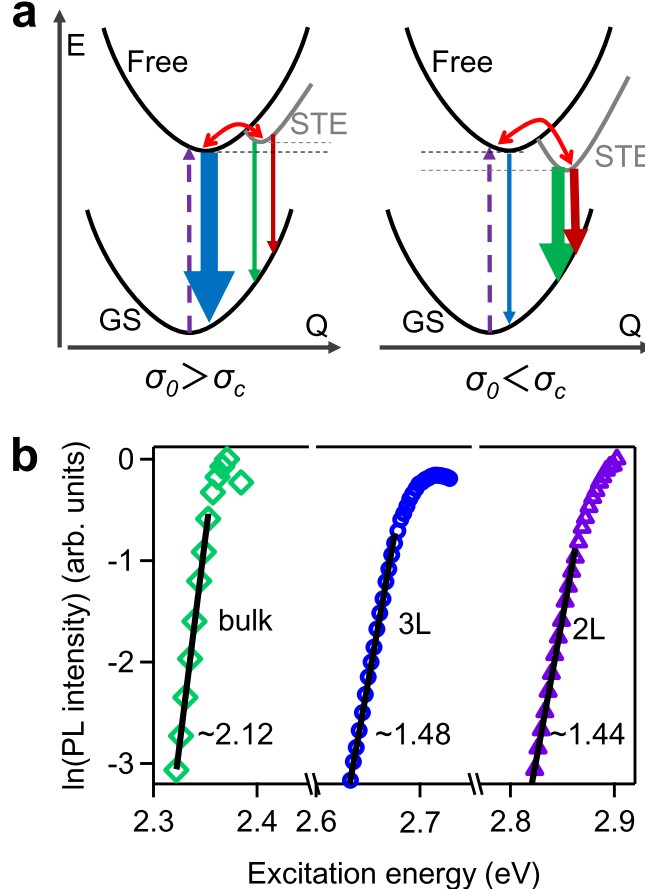

**Fig. 4 Urbach rule in lead halide perovskites. a** Scheme of the energy diagram of free and STE state in different regimes and dominant relaxed species. **b** Urbach tail and fitting according to Eq. 4 of different thickness perovskites.

the U–M tail can be expressed by

$$\alpha(E) = \alpha_0 \times e^{\frac{-\sigma(E_0 - E)}{k_B T}} \qquad (4)$$

where absorption coefficient $\alpha$ is a function of photon energy $E$; $k_B$ is Boltzmann constant, $T$ is temperature, $E_0$ is the converging energy, and $\alpha_0$ is the absorption coefficient at $E = E_0$. The most important parameter $\sigma$ is the steepness coefficient, which, together with $k_B T$, represents the degree of steepness of the absorption tail; $\sigma$ increases with temperature and reaches a constant (i.e. steepness constant $\sigma_0$) at high temperature (e.g. room temperature) (see Supplementary Fig. 3 and Supplementary Note 1). Importantly, $\sigma_0$ (not $\sigma$ which is $T$ dependent) is inherent for each material and inversely proportional to the exciton–phonon coupling strength, with a larger $\sigma_0$ (or steeper tail) corresponding to a weaker exciton–phonon coupling[52]. Therefore, we can infer the nature of the relaxed excited state, either a free state or a STE state, based on the magnitude of $\sigma_0$ relative to the critical value $\sigma_c$ (Supplementary Note 2)[54]. As shown in Fig. 4a, if $\sigma_0$ is apparently larger (smaller) than $\sigma_c$, free state (STE) is stable excited-state species for PL emission. This model has been successfully applied in many systems (e.g. $AgBr_xCl_{1-x}$ (ref. [55]), $PbI_2$ (ref. [38]), pyrene[56], and anthracene[57]) to predict whether exciton self-trapping occurs or not. For 2D perovskites, there have been a few attempts to probe the exciton–phonon interaction from the absorption tail[58,59]. However, the Urbach tail and the degree of steepness can contain contributions from the light-scattering artifacts in optical measurements or extrinsic defects/disorders in samples, which could significantly underestimate $\sigma_0$. Extracting intrinsic $\sigma_0$

representing exciton–phonon interaction requires careful and sensitive measurements (e.g. Fourier-transform photocurrent spectroscopy, PLE spectroscopy) on high-quality samples (e.g. single crystals with high PL QY).

To minimize light-scattering interference and achieve a large dynamic range and high sensitivity, we determined the band edge absorption profile by PLE measurement on $CsPbBr_3$ NP solution with a high PL QY. The PLE spectrum and the fit to the low-energy tail by Eq. 4 for 2L $CsPbBr_3$ NPs are shown in Fig. 4b. The extracted converging energy $E_0$ (~2.88 eV) is very close to the exciton absorption peak (~2.85 eV), suggesting that the U–M tail indeed originates from the exciton–phonon interaction. The steepness constant $\sigma_0 = 1.44$ for 2L $CsPbBr_3$ is a little larger than the critical value $\sigma_c = 1.42$ for the 2D system, which indicates an intermediate exciton–phonon interaction strength and small energy difference between STE and FE states[54]. This agrees well with temperature-dependent PL results showing slightly lower energy of free state than STE with an energy difference of ~34 meV (i.e. STE is metastable). Therefore, excitons are momentarily localized from time to time under thermal fluctuation, which can be called Urbach exciton[52]. Considering ~0.25 ps equilibration time, free state and STE are in thermal equilibrium for emission process. The momentarily localized excitons manifest themselves as an exponential tail on absorption and PL spectrum at room temperature, which is a distinct signature of Urbach exciton[52]. Despite higher STE-state energy than free state, the emission energy of STE is still lower than free state because of configuration displacement. Because of intermediate exciton–phonon coupling in perovskite lattice with dynamic disorder and relatively small lattice configuration change (compared to Holstein-like STE), STE can be directly photoexcited at exponential tail and then establish thermal equilibrium with free state at room temperature, as shown by PL and TA results under low-energy excitation. We note this situation is very similar to that in anthracene crystal where STE is also metastable and STE emission is thermally activated at room temperature[60]. In this regime, excitons are not trapped at low temperature but momentarily trapped at high temperature by the vibrating lattice, opposite to conventional semiconductor defect trapping.

We further examine the dimensionality effect to gain more insights about exciton behavior in 2D perovskites. We measured the PLE spectra of 3L $CsPbBr_3$ NPs and bulk $CsPbBr_3$ single crystal and fitted their U–M tails as shown in Fig. 4b. Compared to $\sigma_0$ of 1.44 in 2L $CsPbBr_3$, $\sigma_0$ increases progressively with thickness, with 1.48 for 3L $CsPbBr_3$ and 2.12 for bulk $CsPbBr_3$. Since $\sigma_0$ is mostly contributed by the short-range interaction for exciton[61], this trend suggests that exciton–phonon interaction especially short-range interaction is enhanced in thin 2D perovskites. Compared to 2L $CsPbBr_3$, the larger $\sigma_0$ in 3L $CsPbBr_3$ is consistent with less asymmetric PL spectrum at room temperature (Supplementary Fig. 4), indicating weaker exciton–phonon coupling. The increasing exciton–phonon coupling from bulk to 3L and 2L perovskites is reasonable since excitons in thinner perovskites are more tightly confined in a more deformable and dynamic lattice, in favor of enhanced short-range exciton–phonon interaction[12,20,62]. This leads to an unique intermediate exciton–phonon coupling regime with momentarily trapped excitons in 2D perovskites, which is different from Fröhlich-like large polaron with dominating long-range interaction and Holstein-like STE with strong short-range interaction. It would be interesting to perform same measurements on 1L lead bromide perovskite, but we failed on preparing the sample using this colloidal method. Previous PL results on $(BA)_2PbBr_4$ single crystal show a more prominent low-energy tail[26], suggesting more STE character with stronger exciton–phonon interaction for lead bromide perovskite. The steepness constant $\sigma_0$ in bulk

$CsPbBr_3$ single crystal is comparable to that in high-quality $CH_3NH_3SnI_3$ thin film ($\sigma_0 = 2.19$)[63] and much larger than the critical value for 3D lattice ($\sigma_c = 1.64$)[54]. This indicates the absence of self-trapping in 3D lead bromide perovskites. Indeed, as expected, large polaron formed by long-range Fröhlich interaction is believed to be dominant excited-state species in 3D lead halide perovskites and responsible for the remarkable optoelectronic properties[9,10]. Extrinsic carrier localization in 3D lead halide perovskites has been inferred from spectral-dependent[31] and power-law[64] PL decay behavior, which is different to intrinsic self-trapping in 2D perovskites discussed here.

The momentarily localized exciton in 2D $CsPbBr_3$ perovskites has strong implications to their optoelectronic properties and applications. First, it helps explain more than an order of magnitude lower exciton diffusion constant in few-layer $CsPbBr_3$ 2D perovskites compared to excitons in monolayer transition metal dichalcogenides and carriers in 3D $CsPbBr_3$ at room temperature[17,65]. While a FE polaron can move fast and coherently, its momentary trapping to STE state can significantly reduce the exciton diffusion constant. Fortunately, the exciton lifetime in 2D perovskite is long enough to guarantee long-range transport over hundreds of nanometers in solar cell devices[17,65]. Second, the thermal equilibrium between free state and STE state which contributes to PL tail is harmful to color purity in light-emitting devices. This calls for effective heat management for high color purity[26]. On the other hand, controlling the inherent exciton–phonon interaction in 2D perovskites becomes important[3,28]. Our results above are based on OLA-capped $CsPbBr_3$ 2D perovskites. Changing surface ligand to, e.g., BA, PEA or halide to I, Cl will likely affect exciton–phonon coupling strength thus exciton behavior, which will be our following study[66]. In fact, ligand engineering to regulate the crystal rigidity has been shown as an effective way to control exciton–phonon coupling to achieve bright blue Pb–Br perovskite emitters[3] and enhanced yellow electroluminescence from 2D Pb–I perovskites[4]. However, we also note that the origin about low-energy broadband emission in BA- or PEA-$PbI_4$ 2D perovskites is still highly debated[33,48,58,66]. Whether it is from intrinsic STE or surface defects requires further careful experimental investigation.

In conclusion, we performed detailed spectroscopy study to reveal the nature of excitons in 2D $CsPbBr_3$ perovskites. We show that the asymmetric PL spectrum with low-energy exponential tail is intrinsic and originates from momentarily trapped exciton polarons with intermediate exciton–phonon interaction strength. The STE is metastable with a small activation energy of ~34 meV and reaches thermal equilibrium with FE polaron in ~0.4 ps, much faster than their recombination process. This study reveals a complex and dynamic picture of exciton polarons in 2D $CsPbBr_3$ and have strong implications to their light-emitting applications. Efforts such as ligand engineering[3,4] and cation doping[4] should be made to regulate exciton–phonon coupling strength for high color purity light-emitting devices.

## Methods

**Sample fabrication.** Two-layer $CsPbBr_3$ perovskites are synthesized according to literature with a slight modification[21]. To a solution of 1.25 mL of 1-octadecene (ODE), 0.125 mL of oleylamine (OLAM) and 0.125 mL of oleic acid (OA), 7 μL of hydrobromic acid (HBr, 48 wt.% in $H_2O$) and 0.1 mL of Cs-oleic acid precursor (0.1 M, 0.35 g of $Cs_2CO_3$ degassed in 20 mL of ODE and 1.25 mL of OA at 150 °C) was swiftly injected 0.2 mL of $PbBr_2$ precursor (0.4 M, 735 mg of $PbBr_2$ in 5 mL of DMF) under stirring. The solution turned turbid white within several seconds, and after 10 s, 5 mL of acetone was swiftly added to quench the reaction. After stirring for an additional 1 min, the NPs were precipitated by centrifugation at 3500 r.p.m. for 5 min and redisposed in 2 mL of toluene. The solution was aged for one night before use. For temperature-dependent measurements, the solution was spin-coated onto quartz substrate; 3L $CsPbBr_3$ perovskite NPs were synthesized by a previous reported solution approach with slight modification. To prepare precursor solution, 367 mg of $PbBr_2$ and 212 mg of CsBr were dissolved in 2 mL of dimethyl

sulfoxide (DMSO) and 1 mL of 40 wt% HBr, respectively. To a mixture of 5 mL toluene, 0.5 mL 1-octadecene, 0.5 mL oleic acid, and 0.5 mL oleylamine, 0.1 mL of $PbBr_2$ precursor and 0.1 mL of CsBr precursor were added successively under continuous stirring. About 20 s later, the solution turned turbid white. At this time, 0.5 mL of butanol was added rapidly to quench the reaction. After 1-min stirring, the NPs were collected by holding the supernatant after centrifugation at 8000 r.p.m. The supernatant was further annealed in oven at 80 °C for 2.5 min. After the supernatant cooled down and turned turbid blue, it was centrifuged again at 10,000 r.p.m. for 10 min to collect the precipitation. The precipitation was dispersed and stored in 2 mL of toluene for further characterization.

**Steady-state optical measurements.** UV-Vis spectrum was taken on Agilent Cary 4000 UV-vis spectrophotometer. PL and PLE spectra of 2L/3L $CsPbBr_3$ in toluene were taken on an Edinburgh Instruments FLS920 spectrometer. The PLE spectrum of $CsPbBr_3$ bulk single crystal was taken on a home-built microscope setup. The emitted photons were collected by photomultiplier and transformed to electrical signal, which is further amplified by pre-amplifier and detected by lock-in amplifier.

**Temperature-dependent measurements.** Temperature-dependent PL and TRPL measurements were taken on a home-built microscope setup. The 380 nm excitation light was generated from a laser system (Light Conversion Pharos, 1030 nm) with optical parametric amplifier. The PL spectra were collected by a spectrograph and analyzed by electron multiplying CCD (ProEM1600 + SP2300, Princeton Instruments). TRPL decay kinetics were collected and analyzed using a TCSPC module (PicoHarp 300) and a SPAD detector (IDQ, id100) with an instrument response function ~100 ps. For temperature-dependent measurements, the sample was placed in a high-vacuum microscope cryostat (MicrostatN Oxford Instruments).

**TA measurement.** For femtosecond TA spectroscopy, the fundamental output from Yb:KGW laser (1030 nm, 100 kHz, Light Conversion Ltd) was separated to multiple light beams. One was introduced to a non-collinear optical parametric amplifier and a second harmonic generation crystal BBO to generate a certain wavelength for pump beam. The other was focused onto a BBO crystal to generate 515 nm pulses and then focused onto a YAG crystal to generation continuum blue light as probe beam. The temporal delay between them is controlled via a motorized delay stage. The pump and probe pulses overlapped spatially in the sample and the transmitted probe light was collected by a linear array detector (TA-100, Time-Tech Spectra, LLC).

## Data availability
The source data necessary to support the findings of this paper are available from the corresponding author upon request.

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

## Acknowledgements

We thank the financial support from the National Natural Science Foundation of China (22022305, 21773208, 21803055), National Key Research and Development Program of China (2017YFA0207700), and the Fundamental Research Funds for the Central Universities.

## Author contributions

W.T. and H.Z. conceived the study; W.T. prepared the samples and performed the optical measurements with the help of C.Z., Q.Z., and Y.Z.. W.T. and H.Z. wrote and revised the manuscript.

## Competing interests

The authors declare no competing interests.
