## [Peer Review File · Nature Communications]

REVIEWER COMMENTS

Reviewer #1 (Remarks to the Author):

In this manuscript, Tao and coauthors report a unique intermediate exciton-phonon coupling regime in two-layer CsPbBr₃. Such exciton polarons gives rise to intrinsic asymmetric photoluminescence with a low energy tail at room temperature and are only momentarily self-trapped by lattice vibrations. The conclusions are supported by static and time-resolved optical spectroscopy and detailed analysis. The manuscript is well organized and this topic is important to this community. I believe that this manuscript can make a contribution to the field, especially for STE related optoelectronic devices. Therefore, I would like to recommend its publication after the authors successfully addressed the following my concerns.

My main concern for this manuscript is that the energy level of self-trapped states is higher than that of free excitons but without providing enough evidence. In view of the weak emission of self-trapped excitons, if the energy level of the self-trapped states is around 35 meV above that of free excitons, the population of self-trapped excitons should be very low. Under such case, we did not expect to observe the self-trapped exciton emission. In addition, the emission spectrum should be equally broadened rather than only exhibit a long energy tail.

It is well known that self-trapped states are with excited state configurational landscape. Under such case, it is impossible to form self-trapped excitons under the excitation energy smaller than that of free excitons. Nevertheless, the authors excited 2L CsPbBr₃ with a low energy pulse (with a center at 2.75 eV and a high energy cutoff at 2.83 eV) to selectively excite the absorption tail and still can observe the long emission tail, suggesting that this long emission tail might not be from the self-trapped states.

For 2L CsPbBr₃, surface-to-volume ratio should very larger. Considering the non-layered nature of CsPbBr₃, a huge amount of surface dangling bond should be present. Under such case, the surface effect such as surface depletion field should be taken into consideration. The surface effect might contribute to the long emission tail as well. The authors are suggested to measure the thickness dependent PLQY to check the surface effect.

Although the 3D perovskite with a diatomic layer can be regarded as a two-dimensional material, there are still differences in structure compared with 2D perovskite we usually investigate (such as (BA)₂PbI₄, (PEA)₂PbI₄), especially the difference of organic chains. In previous reported literature (Nat. Commun. 2019, 10, 806), the slope of (BA)₂(MA)Pb₂I₇ ($\sigma \approx 0.7$) is obviously smaller than that in this manuscript, indicating a stronger exciton-phonon coupling strength. This difference indicates that the organic chain may affect the coupling of excitons and phonons. Therefore, I suggest the authors to clearly illustrate this difference in their manuscript. Another possible reason for their smaller σ is that they did not measure the absorption spectra at different temperatures to fit Urbach tail, resulting in larger fitting errors. In addition, the surface depletion field might also affect the Urbach tail. The authors are suggested to comment this in their manuscript.

Considering that the size of the sample is only 20 nm, which is much smaller than the collection size of a spectrometer, the signals emitted from the edge and surface of the sample both are collected. Given the larger proportion of the edge surface, it is important to consider the influence of the edge state luminescence. If surface/edge states are dominating in the material, it may be difficult to identify them by power-dependent spectrum solely.

Within a certain power range, for example, two orders of magnitude in this manuscript, LE and FE have a similar linear law, which can eliminate the suspicion of external defects. However, previous report reveals that the luminescence peak under the band gap of the two-dimensional perovskite has a different power dependence from the free exciton only under strong excitation light, and the luminescence peak under the band gap is also attributed to STE (Sci. Adv. 2020, 6, eaay4900). Therefore, the range of the excitation intensity should be enlarged to check that.

I donot think it is proper to call such 2L CsPbBr₃ as 2D RP perovskites.

The authors are suggested to provide evidence that their samples are indeed 2L in thickness. More characterizations on morphology and crystalline quality of their samples should be provided.

Some important references are missing in terms of the extraction of exciton-phonon coupling strength in 2D perovskite by Urbach tail (e.g. Nat. Commun. 2019, 10, 806).

There are some typos, e.g. page 9 'Previous studies on 2D lead halide perovskites also have show ...' should be 'shown'.

Reviewer #3 (Remarks to the Author):

In this manuscript, the authors mainly use temperature-dependent PL spectra and TA spectra to investigate the exciton activities in 2D lead halide perovskites, and the momentarily interconversion between free and self-trapped exciton polaron gives rise to intrinsic asymmetric photoluminescence with a low energy tail. There are no new testing techniques, theories, and mechanisms to address the conclusion. However, the result is interesting and valuable to the research of 2D perovskite materials. Before further consideration, the following questions must be fully addressed.

1. There are so many formats, grammatical and typo errors in the article. For example, the footnotes in figure captions of Fig. 1 and Fig. S3 are not in the right form. The figure legend of Fig. 2 has two (d), but no (e). There is no Fig. 4 in the manuscript. Some sentences are hard to understand, the English language of this article should be polished thoroughly, and some language errors should be modified.
2. The reviewer can't see the two layers of CsPbBr₃ NPs in Fig. 1a, the authors should provide clearer TEM images of 2L CsPbBr₃ and 3L CsPbBr₃ to confirm the number of layers.
3. Equ. 4 shows the function of the absorption coefficient with photon energy E. However, why do you use Equ. 4 to fit PLE intensity in Fig. 5b? Is it suitable?
4. Since LE was weak under low temperature, the temperature-dependent steepness constant (σ) should also be plotted and discussed.
5. Why the 2D perovskite has this kind of exciton activity? Does it have some relation with its 2L or 3L structure? The authors should add a discussion in the manuscript.
6. The review notice that the authors use the sentences like "The exact nature will be discussed later" and "As we'll show later", which means the manuscript is not in a good arrangement, the logic of the document should be improved.
7. Are the LE and STE the same thing? If they are the same, please unify them.
8. Could the authors provide the TA spectra at low temperatures? It will help verify and explain why the PL spectra at low temperatures have symmetric shapes.

We thank all reviewers for the generally positive and valuable comments to improve the manuscript. We have performed substantial new experiments and addressed questions and concerns raised by reviewers in the revised manuscript. Following the questions and comments, we made the following corrections and improvements point by point in response. The manuscript has been improved significantly. The responses are in **red** and revisions are in **blue**.

Reviewer #1 (Remarks to the Author):

In this manuscript, Tao and coauthors report a unique intermediate exciton-phonon coupling regime in two-layer CsPbBr₃. Such exciton polarons gives rise to intrinsic asymmetric photoluminescence with a low energy tail at room temperature and are only momentarily self-trapped by lattice vibrations. The conclusions are supported by static and time-resolved optical spectroscopy and detailed analysis. The manuscript is well organized and this topic is important to this community. I believe that this manuscript can make a contribution to the field, especially for STE related optoelectronic devices. Therefore, I would like to recommend its publication after the authors successfully addressed the following my concerns.

Response: We sincerely thank reviewer for the positive comments on this interesting study.

My main concern for this manuscript is that the energy level of self-trapped states is higher than that of free excitons but without providing enough evidence. In view of the weak emission of self-trapped excitons, if the energy level of the self-trapped states is around 35 meV above that of free excitons, the population of self-trapped excitons should be very low. Under such case, we did not expect to observe the self-trapped exciton emission. In addition, the emission spectrum should be equally broadened rather than only exhibit a long energy tail.

Response: We appreciate reviewer for raising the important question about the relative energy level of self-trapped exciton and the free exciton. This is also the key finding of this study which has not been illustrated before. The conclusion that the energy level (NOT the transition energy) of self-trapped states is higher than that of free excitons is mainly evidenced by 1) the temperature-dependent PL spectra where the tail emission is thermally activated at high temperature and 2) further rationalized by the Urbach-Martienssen rule pointing to an intermediate exciton-phonon coupling strength and a metastable STE state with higher energy than free exciton.

We only observe STE emission at room temperature but not at low temperature, opposite to conventional low energy defect PL which is stronger at lower temperature. In the studied 2D perovskites, free-exciton and self-trapped exciton can achieve thermal equilibrium before their radiative/nonradiative recombination at room temperature, which is evidenced by wavelength-independent PL decay kinetics and sub-picosecond conversion observed in TA measurement. According to Boltzmann distribution, neglecting DOS/degeneracy difference, the self-trapped exciton population is estimated to be 28% of free exciton population at room temperature with an energy level difference of 35 meV. Therefore, the STE population is not

that low and could be observed at room temperature PL spectra in this system. On the other hand, if we assume STE state is lower in energy than free exciton, we would expect stronger STE emission than free exciton thus a more asymmetric PL spectra at low temperature, opposite to experimental results.

As can be seen from Figure 4a, the lowest energy configurations of free exciton state and self-trapped exciton state are different, depending on the coupling constant. In our case (left panel), although the STE state is believed to be higher in energy than FE state, the optical transition energy (PL energy) of STE is still lower than that of FE. Therefore, STE emission is much lower and only contribute to the lower energy tail (note the configuration displacement in horizontal axis). In fact, similar low energy emission tail from metastable STE has previously been observed in PbI_2 with $\sigma = 1.48$, close to F-S boundary ($\sigma_c = 1.64$ in 3D) and thermal equilibrium between FE and metastable STE is established before emission process. (Solid State Commun. 56, 101, 1985; Solid State Commun. 59, 209, 1986).

Revisions:

1) page 9, middle, we added

“Based on Boltzmann distribution, LE state population is estimated to be ~ 28% of the FE state population at room temperature, thus emission from LE state can be observed.”

2) page 14, bottom, we added

“Despite of higher STE state energy than free state, the emission from STE is still lower than free state because of configuration displacement.”

It is well known that self-trapped states are with excited state configurational landscape. Under such case, it is impossible to form self-trapped excitons under the excitation energy smaller than that of free excitons. Nevertheless, the authors excited 2L CsPbBr_3 with a low energy pulse (with a center at 2.75 eV and a high energy cutoff at 2.83 eV) to selectively excite the absorption tail and still can observe the long emission tail, suggesting that this long emission tail might not be from the self-trapped states.

Response: We appreciate reviewer for raising the important question about the excitation of self-trapped state. We agree with reviewer that self-trapped states are with excited state configurational landscape. Within the framework of Frank-Condon principle with vertical transition only, the possibility of a transition being directly excited from electronic ground state depends on the relative ground state population at a certain lattice configuration (which is further determined by phonon population and temperature).

For materials such as white light emitting perovskites with strong exciton-phonon interaction where STE state dominates the excited state properties, STE state is accompanied with large lattice configuration displacement compared to FE state (Holstein-like STE, red curve and transition below). In this case, the ground state population at STE configuration is very small at room temperature thus the direct vertical excitation of STE from ground to excited states is negligible.

In our case of 2L perovskites, exciton-phonon coupling strength is in intermediate

regime and PL emission is still dominated by FE emission. The lattice configuration change is not large and the ground state at STE configuration is accessible due to the soft, anharmonic, and dynamically disordered perovskite lattice at room temperature (green curve and transition below). Therefore, the STE transition can be excited at room temperature, as shown by PL (Fig. 2) and TA (Fig. 3) study. By exciting low energy STE transition, both PL and TA studies show thermal equilibrium between FE and STE. We also performed temperature dependent PLE measurement in the revised manuscript which shows reduced absorption tail with reducing temperature, consistent with more symmetric PL at low temperature. Therefore, both steady state and time-resolved PL and absorption measurements point to the same picture.

Indeed, as reviewer pointed out, the absorption corresponding to this STE transition is very weak compared to FE transition, both absorption and PL only show an exponentially decay tail without any distinct peak, corresponding to Urbach tail which has been ascribed to the momentarily localized exciton states (OPTICAL PROCESSES IN SOLIDS, 2003, YUTAKA TOYOZAWA)

Revisions:

1) page 9, bottom, we revised:

“to selectively excite the exponential absorption tail which is dominated by LE exciton or the momentarily exciton transition”

2) page 14, bottom, we added:

“Because of intermediate exciton-phonon coupling in perovskite lattice with dynamic disorder and relatively small lattice configuration displacement (compared to Holstein-like STE), STE can be directly photoexcited at exponential tail and then establish thermal equilibrium with free state at room temperature, as shown by PL and TA results.”

For 2L CsPbBr₃, surface-to-volume ratio should very larger. Considering the non-layered nature of CsPbBr₃, a huge amount of surface dangling bond should be present. Under such case, the surface effect such as surface depletion field should be taken into consideration. The surface effect might contribute to the long emission tail as well. The authors are suggested to measure the thickness dependent PLQY to check the surface effect.

Response: We appreciate reviewer for raising the question on the surface effect.

For 2D perovskites, as reviewer pointed out, surface to volume ratio is large, surface

dangling bond should be present if without ligand passivation. However, for these colloidal CsPbBr₃ plates, the surface atoms such as Br vacancies are well passivated by amine ligands, yielding to a high PL QY (ACS Energy Lett. 2018, 3, 2030–2037). If large amount of surface dangling bonds/defects is present, a distinct stokes-shifted broadband trap emission, instead of near band emission with a low energy exponential tail, would be observed as reported in previous literatures (ACS Energy Lett. 2020, 5, 2149–2155; Nat Commun 11, 2344 ,2020) and more importantly, stronger defect-related tail emission would show up at lower temperature. The absence of stokes-shifted broadband emission at room temperature and less tail emission at lower temperature suggest surface dangling bonds are well passivated and surface trap emission doesn't play a role.

We also measure the PLQY of our 2D perovskites as reviewer suggested using two methods. The PLQY for 2L CsPbBr₃ is estimated to be about 38% with perylene in ethanol solution as the standard reference (QY=94%, J. Phys. Chem. 1968, 72, 9, 3251–3260) and to be about 40% by comparing the measured PL decay rate (0.313 ns⁻¹) with reported radiative rate (0.123 ns⁻¹) (Nano Lett. 2018, 18, 8, 5231–5238). The PLQY of our sample is comparable with the highest reported PLQY (49%) of deep blue emitting perovskite materials (Nano Lett. 2018, 18, 8, 5231–5238), suggesting our 2L CsPbBr₃ NPs in this study are of high quality. As for 3L CsPbBr₃ NPs, the PLQY is estimated about 90%, comparable to the value in the followed literature (96%, ACS Energy Lett. 2018, 3, 2030–2037).

Revisions:

1) Page 5, top, we revised:

“The PL quantum yield (QY) is measured to be ~ 38%, which is comparable with the highest reported PLQY (49%) of 2L CsPbBr₃ and suggests high sample quality.”

2) Page 7, bottom, we revised:

“This is also not expected for surface/edge state emission which was observed previously in 2D perovskites and characterized with lower energy and longer PL lifetime compared to free exciton emission.”

3) Page 9, top, we added:

“This temperature dependent behavior of LE emission cannot be ascribed to surface/edge-related states which should yield more pronounced LE emission at lower temperature.”

Although the 3D perovskite with a diatomic layer can be regarded as a two-dimensional material, there are still differences in structure compared with 2D perovskite we usually investigate (such as (BA)₂PbI₄, (PEA)₂PbI₄), especially the difference of organic chains.

In previous reported literature (Nat. Commun. 2019, 10, 806), the slope of (BA)₂(MA)Pb₂I₇ ($\sigma \approx 0.7$) is obviously smaller than that in this manuscript, indicating a stronger exciton-phonon coupling strength. This difference indicates that the organic chain may affect the coupling of excitons and phonons. Therefore, I suggest the authors to clearly illustrate this difference in their manuscript. Another possible reason for their smaller σ is that they did not measure the absorption spectra at different temperatures to fit Urbach tail, resulting in larger fitting errors. In addition, the surface depletion field might also affect the Urbach tail.

The authors are suggested to comment this in their manuscript.

Response: We appreciate reviewer for raising the question about the organic chain effects on the exciton-phonon coupling strength which indeed is really very interesting and important.

As reviewer pointed out, ligand organic ligands could affect exciton-phonon coupling strength in 2D perovskites. The organic ligands for our CsPbBr₃ is oleylammonium (OLA), which has same functional binding group but is longer than BA or PEA ligands. A recent literature (J. Phys. Chem. Lett. 2020, 11, 20, 8565–8572) show some preliminary data of ligand effect on exciton phonon coupling, which requires further investigation. The ligand effect would be our next step. I am afraid we cannot say too much about this topic now, without any results.

The literature (Nat. Commun. 2019, 10, 806) reviewer mentioned proposed a stronger exciton-phonon coupling strength in (BA)₂(MA)Pb₂I₇ and assigned the low energy PL peak to STE. However, this is still under strong debate. For example, recent two literatures point out that extrinsic defects such as halide vacancies and surface defects are responsible for the broadband emission peak in PEA₂PbI₄ (ACS Energy Lett. 2020, 5, 2149–2155; Nat Commun 11, 2344, 2020). Again, without performing experiments on this sample, we cannot comment too much.

As for the steepness constant for (BA)₂(MA)Pb₂I₇ in Nat. Commun. 2019, 10, 806, the σ_0 is estimated about 0.7 (left panel below), which seems to be too small compared to critical value of 2D case i.e. $\sigma_c = 1.42$. Such low σ_0 would indicate a lowest energy STE state where all excitons should localize to, especially at low temperature. However, their PL spectra show a dominant free exciton emission with only a PL tail (right panel below). As we have stated clearly in revised manuscript, the Urbach tail and the degree of steepness can be easily contaminated by the light scattering in measurements and extrinsic defects/disorders in samples. This can significantly lower σ_0 value and doesn't reflect the true σ_0 due to intrinsic exciton-phonon interaction in a material.

Reproduced from Li, J., Wang, J., Ma, J. et al. Self-trapped state enabled filterless narrowband photodetections in 2D layered perovskite single crystals. *Nat Commun* **10**, 806 (2019). <https://doi.org/10.1038/s41467-019-08768-z>

Revisions:

1) page 4, middle, we added:

“The NPs are capped with oleylammonium ligands and can be well dispersed in nonpolar solvents or deposited into films using drop cast or spin coating. Compared to conventional 2D RP perovskites with butylammonium (BA) or phenylethylammonium (PEA) ligands, we also call these NPs as 2D RP perovskites because of same inorganic part and optical properties. Actually, this kind 2D perovskite NPs can be reversibly assembled into ordered superstructure, i.e. multi-layered

RP perovskite people usually investigated.”

2) page 13, bottom, we added:

“For 2D perovskites, there have been a few attempts to probe the exciton-phonon interaction from the absorption tail. However, the Urbach tail and the degree of steepness can be easily contaminated by the light scattering in measurements and extrinsic defects/disorders in samples, which could significantly underestimate σ_0 . Extracting the intrinsic σ_0 due to exciton-phonon interaction requires a careful measurement on a high-quality sample.”

3) page 17, top, we added:

“Our results above are based on oleylaminium-capped CsPbBr₃ 2D perovskites. Changing surface ligand to e.g. BA, PEA or halide to I, Cl will likely affect exciton-phonon coupling strength thus exciton behavior, which will be our following study.”

4) page 17, middle, we added:

“However, we also note the origin about low energy broadband emission in BA- or PEA-PbI₄ 2D perovskites is still under highly debate. Whether it's from intrinsic STE or surface defects requires further careful experiment investigation.”

Considering that the size of the sample is only 20 nm, which is much smaller than the collection size of a spectrometer, the signals emitted from the edge and surface of the sample both are collected. Given the larger proportion of the edge surface, it is important to consider the influence of the edge state luminescence. If surface/edge states are dominating in the material, it may be difficult to identify them by power-dependent spectrum solely.

Response: We thank reviewer for raising the important question on the influence of edge/surface state luminescence. We agree with reviewer that with a large proportion of the edge/surface in our sample, edge state luminescence could play a role. However, we didn't observe the signature of edge state emission in our samples at both room temperature and low temperature. Previous studies (e.g. ACS Nano 2019, 13, 1635–1644) have shown that the edge state is extrinsic and forms where the adjacent layers partially merge together with the help of water molecular. The edge state is broad with lower energy and shows large stokes shift with longer lifetime compared to intrinsic state, which was not observed. On the other hand, although our sample is only 20 nm, the surface/edge is well passivated during NPs synthesis and a high PL QY (38%) was determined for these sample.

Revisions:

1) Page 5, top, we revised:

“The PL quantum yield (QY) is measured to be ~ 38%, which is comparable with the highest reported PLQY (49%) of 2L CsPbBr₃ and suggests high sample quality.”

2) Page 7, bottom, we added:

“This is also not expected for surface/edge state emission which was observed previously in 2D perovskites and characterized with lower energy and longer PL lifetime compared to free exciton emission.”

3) Page 9, top, we added:

“This temperature dependent behavior of LE emission cannot be ascribed to surface/edge-related

states which should yield more pronounced LE emission at lower temperature.”

Within a certain power range, for example, two orders of magnitude in this manuscript, LE and FE have a similar linear law, which can eliminate the suspicion of external defects. However, previous report reveals that the luminescence peak under the band gap of the two-dimensional perovskite has a different power dependence from the free exciton only under strong excitation light, and the luminescence peak under the band gap is also attributed to STE (Sci. Adv. 2020, 6, eaay4900). Therefore, the range of the excitation intensity should be enlarged to check that.

Response: We thank reviewer pointing this out. Following reviewer’s suggestion, we enlarged the excitation intensity range and the results are shown in manuscript below. Both PL spectra and the ratio between FE and STE emission remain same by varying the excitation power over the range of four orders of magnitude. This result strongly implies that the tail emission is intrinsic to 2D perovskites without saturation, in line with our assignments to the meta-stable STE state emission.

Revisions: We revised the Figure.2b with a much broader excitation power range and revised the content accordingly.

I donot think it is proper to call such 2L CsPbBr₃ as 2D RP perovskites. The authors are suggested to provide evidence that their samples are indeed 2L in thickness. More characterizations on morphology and crystalline quality of their samples should be provided.

Response: We thank reviewer for raising this question. The synthesized 2D perovskites is (100)-oriented and can be viewed as dimensional reduction of 3D perovskite lattice. The (100)-oriented 2D perovskites can be further classified into RP and DJ phase according to the charge of the spacer cation (J. Am. Chem. Soc. 2019, 141, 3, 1171–1190). Since the spacer cation in the synthesized 2D perovskites is single charged (protonated oleylamine), we call them 2D RP perovskites. Actually, this kind 2D perovskite NPs can be reversibly assembled into ordered superstructure, i.e. multi-layered RP perovskite we usually investigate (JACS 2019 141 (33), 13028-13032). This sample can be best viewed as unstacked or exfoliated RP perovskites.

We are sorry that we can not directly see the 2L structure with TEM instrument we can access. Obtaining this kind of high-resolution image is very challenging for perovskites

because of their rapid structural degradation under electron beam exposure (Science 2020, 370, eabb5940). Even by a short time exposure for low resolution imaging, we already observed some local structure damage and Pb formation (the black dots on image). Instead, we rely on optical measurements to characterize the sample thickness and crystalline quality. The relationship between the absorption/emission peak and the layer number has been well established in previous literatures (e.g. Nano Lett. 2018, 18, 8, 5231–5238). Due to quantum confinement effect along vertical direction, the strong and sharp absorption/emission peak of 2D perovskites shows a one-to-one unique and precise correspondence with layer number, which provides a facile and precise way to determine the layer number. The NP we use in this study corresponds to the absorption/emission peak of 2L CsPbBr₃. The absence of any other absorption/emission peaks also suggests that our synthesized NPs are of high monodispersed. The crystalline quality of our samples can also be inferred from the high QY of NPs. (~ 38%)

Revisions:

1) Page 4, middle, we added:

“Compared to conventional 2D RP perovskites with butylammonium (BA) or phenylethylammonium (PEA) ligands, we also call these NPs as 2D RP perovskites because of same inorganic lattice, ligand binding group and optical properties. Actually, this kind 2D perovskite NPs can be reversibly assembled into ordered superstructure, i.e. multi-layered RP perovskite people usually investigated.”

2) Page 4, bottom, we revised:

As these atomically thin perovskite NPs are highly susceptible to electron beam damage, we turn to optical measurements to characterize the exact thickness and homogeneity of NPs, which provides a facile and precise approach.

3) Page 5, top, we added:

The PL quantum yield (QY) is measured to be ~ 38%, which is comparable with the highest reported PLQY (49%) of 2L CsPbBr₃ and suggests high sample quality.

Some important references are missing in terms of the extraction of exciton-phonon coupling strength in 2D perovskite by Urbach tail (e.g. Nat. Commun. 2019, 10, 806).

Response: We are sorry we missed some literatures about the Urbach tail in 2D perovskites in previous manuscript. We thank reviewer pointing out the missing reference (Nat. Commun. 2019, 10, 806). Besides the reference the reviewer mentioned, we also find a literature about the temperature-dependent Urbach tail and the steepness constant in n=1 layered double perovskite (BA)₄AgBiBr₈. We have added both of them in the revised manuscript.

Revisions: We have added two missing references about extracting coupling strength by Urbach tail.

There are some typos, e.g. page 9 'Previous studies on 2D lead halide perovskites also have show ...' should be 'shown'.

Response: We are sorry about the typos and have corrected them in the revised manuscript.

Reviewer #3 (Remarks to the Author):

In this manuscript, the authors mainly use temperature-dependent PL spectra and TA spectra to investigate the exciton activities in 2D lead halide perovskites, and the momentarily interconversion between free and self-trapped exciton polaron gives rise to intrinsic asymmetric photoluminescence with a low energy tail. There are no new testing techniques, theories, and mechanisms to address the conclusion. However, the result is interesting and valuable to the research of 2D perovskite materials. Before further consideration, the following questions must be fully addressed.

Response:

We appreciate reviewer for the positive comments on the value of the results to the of 2D perovskites materials. The knowledge of exciton nature in 2D lead halide perovskites is of great importance but significantly lags behind the booming developments of 2D perovskites based optoelectronic devices. Compared with the well-established consensus that large polarons form in 3D lead-halide perovskites through the long-range Frohlich interaction and self-trapped excitons form in white light emitting 2D perovskites with corrugated lattice through short-range Holstein-like interactions, the exciton nature in 2D lead halide perovskite has not been addressed. The manuscript aims to give a new and unified picture of the exciton polaron in 2D lead halide perovskite and reveals an intermedium exciton-phonon coupling with momentarily trapped exciton polaron. Also we reveal the intrinsic origin about the exponential low energy PL tail, which has been under strong debate before.

As pointed out by the reviewer, we probed the effects of exciton-phonon interaction on the nature of excitons in 2D lead halide perovskites with conventional static and time-resolved optical spectroscopies. While these spectroscopic techniques are not new, the combination of them, together with in-depth analysis through Urbach-Martienssen rule, can provide rich information and new picture about the exciton-phonon interaction and exciton nature in 2D lead halide perovskites.

1. There are so many formats, grammatical and typo errors in the article. For example, the footnotes in figure captions of Fig. 1 and Fig. S3 are not in the right form. The figure legend of Fig. 2 has two (d), but no (e). There is no Fig. 4 in the manuscript. Some sentences are hard to understand, the English language of this article should be polished thoroughly, and some language errors should be modified.

Response: We are sorry about typos in manuscript and have corrected them in the revised manuscript. Thank you so much!

2. The reviewer can't see the two layers of CsPbBr₃ NPs in Fig. 1a, the authors should provide clearer TEM images of 2L CsPbBr₃ and 3L CsPbBr₃ to confirm the number of layers.

Response: We are sorry that we cannot directly see the 2L structure with TEM instrument we can access. Obtaining this kind of high-resolution image is very challenging for perovskites because of their rapid structural degradation under electron beam exposure (Science 2020, 370, eabb5940). Even by a short time exposure for low resolution imaging, we already

observed the structure damage and Pb formation. Therefore, instead, we rely on optical measurements to characterize the exact sample thickness and crystalline quality. The relationship between the absorption/emission peak and the layer number has been well established in previous literatures (e.g. Nano Lett. 2018, 18, 8, 5231–5238). Due to quantum confinement effect along vertical direction, the strong and sharp absorption/emission peak of 2D perovskites shows a one-to-one unique and precise correspondence with layer number, which provides a facile and accurate way to determine the layer number. The NP we use in this study corresponds to the absorption/emission peak of 2L/3L CsPbBr₃. The absence of any other absorption/emission peaks also suggests that our synthesized NPs are of high monodispersed.

Revisions:

Page 4, bottom, we revised:

As these atomically thin perovskite NPs are highly susceptible to electron beam damage, we turn to optical measurements to characterize the exact thickness and homogeneity of NPs, which provides a facile and precise approach.

3. Equ. 4 shows the function of the absorption coefficient with photon energy E. However, why do you use Equ. 4 to fit PLE intensity in Fig. 5b? Is it suitable?

Response: We apologize that we didn't make it clear in previous manuscript. As pointed by the reviewer, Equ.4 shows the function of the absorption coefficient with photon energy E. For NPs solution or thin film, light scattering/reflection, instead of real absorption, is hard to avoid, which could hinder the precise measurement of the absorption tail. To minimize these artifacts, we extract the photon-energy dependent absorption tail using PLE spectrum which is proportional to absorption spectrum but has much higher sensitivity and dynamic range. Basically, PLE is a highly sensitive absorption measurement.

As shown by $PLE(\lambda) = N_{\text{photon}}Abs(\lambda)QY(\lambda)$, PLE intensity is given by photon number N at a specific excitation wavelength (λ) which is normalized in PLE measurement, the absorption (in percentage) at λ and quantum yield (QY) at λ which can be assumed to be same within such narrow wavelength range. Therefore, PLE profile directly yields absorption profile.

Revisions:

in page 14, top, we revised

“To minimize light scattering interference and achieve a large dynamic range and high sensitivity, we determined the band edge absorption profile by PLE measurement on CsPbBr₃ NP solution sample with a high PL QY.”

4. Since LE was weak under low temperature, the temperature-dependent steepness constant (σ) should also be plotted and discussed.

Response: We thank reviewer for raising the question about the temperature-dependent steepness constant. Following reviewer's suggestion, we have measured the temperature dependent PLE spectra and extracted the steepness coefficient. The results are shown in Fig. S3 and below. As expected, the degree of steepness increases with temperature (left panel),

from which we extracted steepness coefficient (σ). As shown in right panel, σ increases with temperature and reaches a constant, i.e. steepness constant σ_0 , at room temperature. It is important to note that only the high temperature steepness constant σ_0 represents the intrinsic exciton-phonon coupling constant by Equation S6. All the following discussion about exciton-phonon coupling is based on steepness constant σ_0 at room temperature.

Revisions:

1) we added Fig. S3 and added the associated discussion about it in Supplementary Note 1.

2) page 13, middle, we revised

“The most important parameter σ is the steepness coefficient, which, together with $k_B T$, represents the degree of steepness of the absorption tail. σ increases with temperature and reaches a constant, i.e. steepness constant σ_0 , at room temperature (see Fig. S3). Importantly, σ_0 is inherent for each material and inversely proportional to the exciton-phonon coupling strength (see Supplementary Note 1), with a larger σ_0 (steeper tail) corresponding to a weaker exciton-phonon coupling.”

5. Why the 2D perovskite has this kind of exciton activity? Does it have some relation with its 2L or 3L structure? The authors should add a discussion in the manuscript.

Response: We thank reviewer for raising this interesting fundamental question. As pointed by reviewer, the exciton property should correlate with lattice structure. In principle, exciton activity is determined by exciton-phonon interaction, which is further determined by intrinsic lattice property. In fact, the results of dimensionality effect (comparing 2L, 3L and bulk) provides a clue about this question. As shown by steepness constant, compared to 3D perovskites, short-range exciton-phonon interaction is significantly enhanced in thinner 2D lattice where excitons are tightly confined in a more deformable and dynamic lattice, leading to an intermediate exciton phonon coupling. This unique intermediate exciton-phonon coupling yields a momentarily trapped exciton in 2D perovskites, which is different from Fröhlich-like large polaron with dominating long-range interaction and Holstein-like STE with strong short-range interaction.

Revisions:

in page 15, bottom, we revised

“The increasing exciton-phonon coupling from bulk to 3L and 2L perovskites is reasonable since

excitons in thinner perovskites are more tightly confined in a more deformable and dynamic lattice, in favor of enhanced short-range exciton-phonon interaction.12, 20, 59 This leads to an unique intermediate exciton-phonon coupling regime with momentarily trapped excitons in 2D perovskites, which is different from Fröhlich-like large polaron with dominating long-range interaction and Holstein-like STE with strong short-range interaction.”

6. The review notice that the authors use the sentences like “The exact nature will be discussed later” and “As we’ll show later”, which means the manuscript is not in a good arrangement, the logic of the document should be improved.

Response : We thank review point this problem out. We have revised the manuscript to make it easier to follow. Thanks again!

7. Are the LE and STE the same thing? If they are the same, please unify them.

Response : We apologize that we didn’t make it clear in the manuscript. LE (localized exciton) is a general class of localized exciton suffering localization effects, including intrinsic localization due to exciton-phonon interaction (self-trapped exciton) and extrinsic localization due to defects (trapped exciton). In the beginning, when we discussed the emission spectrum, we observe a low energy tail and at that time, without knowing exact nature, we simply attribute it to LE emission rather than STE emission. Only with more investigation and results, we can reach the conclusion that the LE emission is due to STE and change the notation. We revised the manuscript substantially to make it easier to follow.

Revisions for Q6 and Q7:

1) we removed the sentence with “The exact nature will be discussed later” and “As we’ll show later” and revised content accordingly.

2) **In page 5, bottom, added:**

“The exact origin about the LE emission, e.g. whether it’s intrinsic or extrinsic, is unclear yet.”

3) **in page 8, middle, we revised:**

“The intrinsic nature of the asymmetric PL with a low energy tail can be attributed the polaronic effect in lead halide perovskites and we assign the LE emission to STE emission.”

8. Could the authors provide the TA spectra at low temperatures? It will help verify and explain why the PL spectra at low temperatures have symmetric shapes.

Response: We thank reviewer for the suggestion. Following reviewer’s suggestion, we measured the TA spectra at low temperature and TA spectra comparison between 80K and 300K are shown below. They show similar features characterized with a dominating ground state bleach (GSB) and two photo-induced absorption (PIA) signals on its wings. The low temperature TA spectra does look narrower and more symmetric compared to that at room temperature, consistent with PL result.

REVIEWER COMMENTS

Reviewer #1 (Remarks to the Author):

The authors have significantly improved their manuscript according to the reviewers' comment. Therefore, I would like to recommend its publication after addressing the following minors.

1) The authors claimed that 'the self-trapped exciton population is estimated to be 28% of free exciton population at room temperature with an energy level difference of 35 meV. Therefore, the STE population is not that low and could be observed at room temperature PL spectra in this system. On the other hand, if we assume STE state is lower in energy than free exciton, we would expect stronger STE emission than free exciton thus a more asymmetric PL spectra at low temperature, opposite to experimental results.' Here the authors assumed that the radiative recombination rate of STE is the same or similar to that of free excitons. Usually, this is not the case. The authors are suggested to validate this.

2) The authors claimed that 'the Urbach tail and the degree of steepness can be easily contaminated by the light scattering in measurements and extrinsic defects/disorders in samples. This can significantly lower σ_0 value and doesn't reflect the true σ_0 due to intrinsic exciton-phonon interaction in a material.'. Can the authors provide a method to determine the intrinsic exciton-phonon interaction of a material? Can the authors illustrate more on the claim 'It is important to note that only the high temperature steepness constant σ_0 represents the intrinsic exciton-phonon coupling constant by Equation S6.'? Do those two claims contradict with each other?

Reviewer #3 (Remarks to the Author):

In this revision, the authors have tried to address all the questions; however, there are still several important issues that have not been resolved. The issues are given in the following:

1. There are still uncertain and inaccurate data or statements described in the current revision. In particular, the reviewer shares the same concern with other reviewers that the authors haven't provided enough evidence to prove that their samples are indeed 2L in thickness. The topic discussed in this manuscript is the optical properties of 2L, 3L, and bulk perovskites. If the layer number is unclear, then the accuracy of the result will also be affected. At the same time, a thorough check on the reference (Nano Lett. 2018, 18, 8, 5231–5238) is performed, in which the research team there have compared with each spectrum and discussed the difference among them. Nevertheless, in this work here, the authors just provide a single spectrum with no obvious sign to know the layer number is 2 or 3. Also, the absorbance spectra of 3L and bulk materials are not provided, while the difference of the spectra is also not discussed. In this regard, it is not precise and accurate to determine the layer number with the current approach. More convective characterizations on the morphology and crystalline quality of their samples are essential.

2. The authors said that "absorption and QY at λ which can be assumed to be the same within such narrow wavelength range." However, the reviewer thinks 2.3~2.4 eV, or 2.6~2.7 eV are not narrow wavelength range, the absorption and fluorescence will change dramatically. It is not the typical or classical way to fit the absorption coefficient utilizing the current method.

3. The effect of the luminescence from defects and edges is not fully considered and still unclear in the current revision.

We thank all reviewers for the generally positive and valuable comments to improve the manuscript. We have addressed questions and concerns raised by reviewers in the revised manuscript. The manuscript has been improved significantly. The responses are in **red** and revisions are in **blue**.

Reviewer #1 (Remarks to the Author):

The authors have significantly improved their manuscript according to the reviewers' comment. Therefore, I would like to recommend its publication after addressing the following minors.

Response: We sincerely thank reviewer for the positive comments on our revised manuscript.

1) The authors claimed that 'the self-trapped exciton population is estimated to be 28% of free exciton population at room temperature with an energy level difference of 35 meV. Therefore, the STE population is not that low and could be observed at room temperature PL spectra in this system. On the other hand, if we assume STE state is lower in energy than free exciton, we would expect stronger STE emission than free exciton thus a more asymmetric PL spectra at low temperature, opposite to experimental results.' Here the authors assumed that the radiative recombination rate of STE is the same or similar to that of free excitons. Usually, this is not the case. The authors are suggested to validate this.

Response: We appreciate reviewer for raising the important question about the radiative recombination rate difference between STE and FE. We completely agree with reviewer that the radiative recombination rate should be different for STE and FE. Unfortunately, we cannot

quantify them by directly measuring them. They show same PL decay kinetics (Fig. 2c) due to fast thermal equilibrium between them.

Under quasi-thermal equilibrium condition, the LE/FE emission intensity ratio obeys the equation (3) in the main text, the full mathematical derivation can be found in previous literature discussing the ratio between STE and FE emission intensity in white-light emitting perovskites at a given temperature (Chem. Sci.,2017, 8, 4497–4504). Here we follow their strategy to estimate the relative energy level between STE state and FE state. This kind phenomenological model gives us a simplified description about the energy landscape of 2D RP perovskites.

$$\ln \frac{I_{LE}}{I_{FE}} \propto \ln \frac{k_{r,LE}}{k_{r,FE}} - \frac{\Delta E}{k_B T} \quad (3)$$

The key is that the absolute fraction of STE doesn't matter. Instead, the temperature dependence matters. Less STE emission from FE at lower temperature indicates higher STE energy.

Revisions: Page 9, middle, we added

“We note the relative PL intensity is also affected by the radiative rate constant, which unfortunately cannot be easily determined since they show identical PL decay kinetics under thermal equilibrium (Fig. 2c).”

2) The authors claimed that 'the Urbach tail and the degree of steepness can be easily contaminated by the light scattering in measurements and extrinsic defects/disorders in samples. This can significantly lower σ_0 value and doesn't reflect the true σ_0 due to intrinsic exciton-phonon interaction in a material.'. Can the authors provide a method to determine the intrinsic exciton-phonon interaction of a material? Can the authors illustrate more on the claim 'It is important to note that only the high temperature steepness constant σ_0 represents the intrinsic exciton-phonon coupling constant by Equation S6.'? Do those two claims contradict with each other?

Response: We appreciate reviewer raising the important question on how to determine the intrinsic exciton-phonon interaction of a material. We are sorry we don't make it clear in last response. First of all, both intrinsic electron-phonon interaction and extrinsic defects/disorder can contribute the Urbach tail. Measurements performed on high quality single crystalline sample with less defects or disorder provide more reliable exciton-phonon interaction of a materials. Another important issue is how to avoid measurement artifacts. The Urbach tail is characterized with relatively low absorption coefficient and spans several orders of magnitude, which is hard to determine from direct absorption measurement, especially for bulk semiconductors where the scattering or reflection contributions are not negligible. Instead, indirect measurements such as Fourier-transform photocurrent spectroscopy (FTPS) and photoluminescence excitation spectrum (PLE) can provided more accurate measurements on the absorption tail. By combining both high quality sample with sensitive measurements, the Urbach tail due to exciton-phonon coupling can be more reliably determined.

We are sorry we didn't make it early about difference between steepness coefficient σ and steepness constant σ_0 in last revision. This is about how to define steepness constant. The steepness of the Urbach tail for a particular material are temperature dependent. The steepness coefficient would approach a constant value when temperature is sufficiently high, e.g. at room temperature. Only this steepness constant σ_0 inversely correctly with exciton-phonon coupling constant and is the intrinsic property of a material. The key is to differentiate between steepness coefficient σ and steepness constant σ_0 .

Revisions:

Page 14, middle, we revised

“However, the Urbach tail and the degree of steepness can contain contributions from the light scattering artifacts in optical measurements or extrinsic defects/disorders in samples, which could significantly underestimate σ_0 . Extracting intrinsic σ_0 representing exciton-phonon interaction requires careful and sensitive measurements (e.g. Fourier-transform photocurrent spectroscopy, PLE spectroscopy) on high-quality samples (e.g. single crystals with high PL QY).”

Page 13, bottom, we revised

“represents the degree of steepness of the absorption tail. σ increases with temperature and reaches a constant (i.e. steepness constant σ_0) at high temperature (e.g. room temperature) (see Fig. S3 and Supplementary Note 1). Importantly, σ_0 (not σ which is T dependent) is inherent for each material”

Reviewer #3 (Remarks to the Author):

In this revision, the authors have tried to address all the questions; however, there are still several important issues that have not been resolved.

We sincerely thank reviewer for the careful reading and constructive comments which help to improve the manuscript significantly.

The issues are given in the following:

1. There are still uncertain and inaccurate data or statements described in the current revision. In particular, the reviewer shares the same concern with other reviewers that the authors haven't provided enough evidence to prove that their samples are indeed 2L in thickness. The topic discussed in this manuscript is the optical properties of 2L, 3L, and bulk perovskites. If the layer number is unclear, then the accuracy of the result will also be affected. At the same time, a thorough check on the reference (Nano Lett. 2018, 18, 8, 5231–5238) is performed, in which the research team there have

compared with each spectrum and discussed the difference among them. Nevertheless, in this work here, the authors just provide a single spectrum with no obvious sign to know the layer number is 2 or 3. Also, the absorbance spectra of 3L and bulk materials are not provided, while the difference of the spectra is also not discussed. In this regard, it is not precise and accurate to determine the layer number with the current approach. More convictive characterizations on the morphology and crystalline quality of their samples are essential.

Response: We appreciate reviewer for the suggestion to provide more evidence to prove our sample thickness. We completely understand reviewer's concern about the layer thickness. Indeed, the layer thickness is very important.

In the revised manuscript, following reviewer's suggestion, we plotted the absorption and emission spectra of our 2L and 3L samples together and also compared with literature results systematically to distinguish their thickness (Fig. S4 and below). It can be seen that the absorption/emission peaks of 2L and 3L NPs are very sharp and well-separated (by 26 nm) due to quantum confinement and excitonic effect. The absence of any other absorption/emission peaks (below, left panel) and identical peak positions with the literature (below, right panel) confirm our NPs are indeed in 2L or 3L in thickness and of high thickness homogeneity. Because of such integer layer numbers and distinct and well-separated (~ 26 nm) excitonic features, the layer number of few layer 2D perovskites can be safely determined by optical approaches.

[REDACTED]

Revisions: We added the absorption/emission spectra of 2 ML and 3 ML NPs and associated discussion for careful comparison in Fig. S4 as reviewer suggested.

2. The authors said that “absorption and QY at λ which can be assumed to be the same within such narrow wavelength range.” However, the reviewer thinks 2.3~2.4 eV, or 2.6~2.7 eV are not narrow wavelength range, the absorption and fluorescence will change dramatically. It is not the typical or classical way to fit the absorption coefficient utilizing the current method.

Response: We are sorry that we didn't make it clearly in previous submission. The review is absolute right the absorption and absolute

PL intensity change with energy. It's not flat but described with an exponential tail. Instead, we were to say the PL QY at different excitation energy in such narrow energy range where we fit Urbach tail (2.82-2.87 eV for 2L, 2.63-2.68 eV for 3L, 2.32-2.35 eV for bulk) can be assumed to be a constant. As we shown by TA, they establish equilibrium very quickly, a few orders of magnitude faster than PL decay. As a result, photoluminescence excitation spectrum reflects absorption profile.

The conventional absorption spectra can be measured with just typical transmission approach. But for low energy Urbach tail with very small intensity, it can be easily contaminated by scattering artifacts in solid state samples. Alternatively, PLE provides a much more sensitive and reliable approach.

To further justify this method, we compared the absorption tail measured by both typical absorption spectrum and PLE spectrum for our NP solution sample where the scattering and reflection artifacts are negligible. As shown below, they yield same Urbach tail. The excellent agreement of absorption spectrum and PLE spectrum also confirms the PL QY changes negligibly in this narrow energy range. Otherwise the PLE spectrum would diverge from the exponential function significantly. This comparison is performed on highly dispersed solution sample. However, for solid state samples (e.g. bulk semiconductors or films) where reflection and scattering contributions are not negligible, methods such as PLE spectrum or Fourier-transform photocurrent spectroscopy (FTPS) can provide more precise way to determine the low energy absorption tail.

3. The effect of the luminescence from defects and edges is not fully considered and still unclear in the current revision.

Response: We appreciate reviewer for raising the important question about the effect of the luminescence from defects and edges. We agree with reviewer that distinguishing these extrinsic effects from intrinsic self-trapped exciton emission is very important to draw a correct conclusion. As reviewer suggested, we added some discussion in previous submission and add more discussion in this revised manuscript.

We distinguish the self-trapped exciton emission and defects/edges emission by several measurements. If the luminescence from defects and edges contributes substantially or dominates the emission tail, it should also manifest in temperature dependent PL spectra and kinetics. However, our experimental results are opposite to those expected for defects or edges.

Edge state and defects emission has been extensively studied in previous literatures. Layer edge state (LES) was first observed and proposed by Mohite group (Science 355, 1288–1292 (2017) and they claimed that the LES is characterized with lower energy and longer PL lifetime compared with higher-energy X states. They argued that the LES state is an intrinsic free carrier like state lower in energy

compared with photogenerated excitonic states which provides an efficient exciton dissociation pathway in RP perovskites. The LES also exhibits saturation behavior at high excitation density as they reported. After one year, Zhang group also reported effective exciton funneling into layer edges and efficient exciton dissociation at edge states in layered-perovskite nanowires. (Nature Electronics 1, 404–410, 2018) It worth noting that both reports suggest that edges states are substantial only when the layer number is larger than 2.

Owing to its importance, there are many following research works aiming to address the nature of the LES. For example, Sargent group suggests that the edge state is due to the stochastic loss of PEA and formation of bulk CsPbBr₃ perovskites which can be stabilized by phosphine oxide molecules. (Nature Communications ,11, 170 ,2020) In their confocal time-resolved PL decay measurements, the edge state lifetime is very sensitive to TPPO treatment while band edge emission from crystal center is rather insensitive. The edge state lifetime is substantial longer than band edge state. (below, left panel) Dou group, Jin group and Bao group also studied the nature of the edge state independently, they both suggest that the edge states in 2D perovskite are extrinsic and can be reversibly controlled by atmospheres, chemical treatments or external hydrostatic pressure. (ACS Nano 2019, 13, 1635–1644; J. Phys. Chem. Lett. 2019, 10, 3950–3954; Chem. Mater. 2020, 32, 5009–5015) It worth noting that the edge state emission is very pronounced at cryogenic temperature in the work reported by Dou etc. (below right panel)

[REDACTED]

There are also many reports discuss the trap state in 2D perovskites. Omar F. Mohammed group reported I vacancies in $(\text{PEA})_2\text{PbI}_4$ would induce stokes-shift broadband emission which can be passivated by PEAI treatment. The lifetime of defect emission is also longer than band edge emission. (below, left panel, ACS Energy Lett. 2020, 5, 2149–2155)

Maria A. Loi group also reported that broad emission related with defect sites. The broad emission is very pronounced at low temperature. (below, right panel, Nature Communications 11, 2344 ,2020)

Reproduced from Kahmann, S., Tekelenburg, E.K., Duim, H. et al. Extrinsic nature of the broad photoluminescence in lead iodide-based Ruddlesden–Popper perovskites. *Nat Commun* **11**, 2344 (2020). <https://doi.org/10.1038/s41467-020-15970-x>

In a word, the layer-edge state and trap state emission can be summarized as follows: they are both extrinsic due to crystal imperfection and can be reversibly controlled. They are both located lower energy and characterized with longer lifetime comparing with band edge exciton. At lower temperature, the layer-edge state/trap state emissions become more pronounced.

While the layer-edge state can also contribute the lower-energy

tail in 2D perovskite in principle, the emission wavelength independent PL kinetics and more symmetric PL spectra at lower temperature in our case cannot be explained by neither layer-edge state nor trap state. As we claimed and discussed in the main text, these behaviors can be well explained by the existence of a meta-stable STE state.

Revisions: on page 10, bottom, we revised

“Previous studies on 2D lead halide perovskites also have shown asymmetric PL line shape and attributed to extrinsic below gap trap states at surfaces/interfaces.^{33,}

⁴³Layer edge states can also contribute the lower emission due the stochastic loss of organic ligands and formation of bulk CsPbBr₃ perovskites.^{2, 32, 44-47} However, lower energy emissions from below gap states or layer edge states exhibit longer lifetime compared with the band edge emission from 2D perovskites and persist at cryogenic temperature, if not be more pronounced.^{2, 32-33, 46, 48} Here in 2L CsPbBr₃ perovskite, the power-independent spectral shape and ultrafast thermal equilibration with a ΔE of ~ 34 meV preclude extrinsic origins such as middle gap states from defects or potential fluctuations from thickness or phase inhomogeneity or unintentionally formed edge states. Otherwise, the low energy tail extending to ~ 450 meV below main peak would indicate a wide distribution of deep trap states or edge states with energy dependent recombination rates due to localization effect and excitons will be localized to these states at low temperature,³³ opposite to experimental results. The absence of extrinsic edge/defect related emission confirms high sample quality, which is key to reveal the intrinsic optical properties.”

REVIEWERS' COMMENTS

Reviewer #1 (Remarks to the Author):

I have no further question and thus would like to recommend its publication at the current form.

Reviewer #3 (Remarks to the Author):

In this revision, the authors have addressed the questions properly. In this regard, it is now ready to accept the manuscript in the current format.